# The Level of Knowledge, Attitudes, and Practices in a Caribbean Colombian Population That Recovered from COVID-19 during the Pandemic

**DOI:** 10.3390/healthcare11243119

**Published:** 2023-12-08

**Authors:** Mirary Mantilla-Morrón, Damaris Suárez-Palacio, Miguel Alberto Urina-Triana

**Affiliations:** 1Physiotherapy Program, Faculty of Health Sciences, Universidad Simón Bolívar, Barranquilla 080002, Colombia; damaris.suarez@unisimon.edu.co; 2Cardiology Fellowship Program, Faculty of Health Sciences, Universidad Simón Bolívar, Barranquilla 080002, Colombia; miguel.urina@unisimon.edu.co

**Keywords:** COVID-19 practices, health emergencies, knowledge level, pandemic

## Abstract

This study aimed to assess the knowledge, attitudes, and behaviors related to COVID-19 among Colombians. A cross-sectional descriptive study was carried out during the pandemic between November 2020 and May 2021 using a survey that focused on knowledge, attitudes, and practices regarding COVID-19. The online survey was completed by 1424 participants who had a history of COVID-19 illness, and the study spanned 3 months. Many respondents were male students who demonstrated adequate knowledge of COVID-19 symptoms and prevention measures, although their understanding of its transmission routes was limited. Nevertheless, 65.9% expressed optimism that COVID-19 would ultimately be successfully controlled, and 71.8% had confidence in the Colombian government’s handling of the crisis. Additionally, more than half of the participants admitted to visiting crowded places, and this practice was more common among those who were less informed about COVID-19. However, many respondents reported using face masks in public. This highlights a significant gap between theoretical knowledge and actual practices that need to be addressed. To bridge this gap, establishing an interdisciplinary support network is crucial, as is launching mass education campaigns targeting specific population groups, and compiling reports on successful practices implemented during the pandemic. These efforts are essential for enhancing the overall level of knowledge, and the attitudes and practices related to COVID-19, and also for preparing for future health emergencies.

## 1. Introduction

COVID-19 is a severe infection that spread rapidly worldwide in 2019, ultimately being declared a pandemic by the World Health Organization (WHO) in March 2020 [1]. In response, many countries implemented strict measures, including lockdowns and economic and social restrictions, to curb the escalating infection rates. However, the cessation of mandatory confinement led to a surge in reinfections, making it increasingly challenging to track and quantify new COVID-19 cases and fatalities [2]. Furthermore, multiple variants of the virus have emerged, some of which are associated with heightened transmissibility or virulence, posing significant concerns, particularly in low-income countries that have yet to complete their vaccination campaigns [2].

The effectiveness of non-pharmaceutical control measures, such as social distancing, handwashing, mask-wearing in public spaces, and the monitoring and self-isolation of suspected cases, remains a subject of debate. The available evidence primarily stems from high-income countries [3]. Implementing these control measures was challenging, as compliance relied heavily on the level of knowledge, attitudes, and preventive practices (KAP) of citizens, and necessitated substantial lifestyle and behavioral changes. These measures were also influenced by historical structural determinants and socioeconomic disparities, impacting their efficacy, and causing psychological and socioeconomic burdens, ultimately leading to the relaxation of emergency health measures [3,4].

Pandemics affect the health and behavior of people, health systems, and innovation. They reveal the importance of prevention, control, balance, and social engagement, and create awareness and improve education about caring for life and nature. Managing and overcoming a global crisis of this magnitude requires knowledge of the disease, public awareness, and compliance with recommendations. The experiences of previous pandemics such as A/H1N1, Zika, and Chikungunya, while valuable, proved insufficient in addressing the scale and persistence of COVID-19, contributing to increased social complexity marked by insecurity and uncertainty [5,6].

In managing and overcoming a global crisis of this magnitude, knowledge of the disease, public awareness, and adherence to recommendations are indispensable. Lessons learned from prior pandemics like A/H1N1, Zika, and Chikungunya, while valuable, proved insufficient in addressing the scale and persistence of COVID-19, contributing to increased social complexity marked by insecurity and uncertainty [5]. 

Various studies have been conducted to evaluate COVID-19 KAPs in different regions and countries around the world. A systematic review and meta-analysis of 84 studies from 45 countries from all continents and found that 75% of people knew about COVID-19, 74% had a good attitude, and 70% did the right thing. However, there were differences according to income level, age, and education. Practice scores were lower in Africa and Europe/Oceania [7]. A cross-sectional study of 10,551 participants from 23 countries in Asia, Africa, Europe, and the Americas revealed that the level of knowledge, attitudes, and practices towards COVID-19 were moderate, with an average score of 13.2 out of 21, 3.6 out of 5, and 7.7 out of 10, respectively [8].

Studies conducted in China and Ethiopia, respectively, revealed that most respondents had good knowledge regarding COVID-19, but their level of preventive practices was relatively low [9,10].

Regarding Latin America, studies conducted in Nicaragua and Cuba, respectively, demonstrated that respondents were aware of the symptoms, transmission, and preventive measures of COVID-19, but only 54% wore masks and 38% practiced social distancing. In Cuba, 97% of the participants had a high level of knowledge, a favorable attitude towards control measures, and adequate prevention practices. However, some knowledge gaps were identified regarding the incubation period, the correct use of masks, and the risk of contagion [11,12].

The KAP framework breaks down the process of behavioral change into three stages: knowledge acquisition, attitude/belief formation, and the adoption of practices/behaviors. This model suggests that behaviors are shaped by an individual’s attitudes and knowledge, enabling active engagement in maintaining and safeguarding one’s health. Therefore, this model plays a vital role in disease prevention, control, and rehabilitation [13].

An individual’s ability to comprehend and utilize information, based on beliefs forged through various sources such as personal experiences, medical guidance, research, the internet, social networks, and factors like causes, exacerbating risk factors, symptom recognition, available treatments, and disease consequences before and after diagnosis, significantly impacts their ability to manage their health condition. While some beliefs may be accurate, others may not be, making having a comprehensive understanding of one’s illness a crucial cognitive factor that influences adaptation [14].

To gain comprehensive insight into a population’s stance on a specific topic, assessing their knowledge level, disposition to think, feel, and act in response to that topic, and understanding the interplay of their beliefs, emotions, and values are essential. Research has consistently demonstrated that protective behaviors, knowledge, attitudes, and risk perceptions are influential factors in preventing and controlling infectious diseases [15].

Researchers have also highlighted the correlation between patients’ KAP levels and the effective management of their diseases, their response to medical treatments, and their ability to promote their overall health [16]. Conversely, a low KAP level serves as an early indicator of poor health outcomes, inefficient healthcare utilization, reduced disease detection rates, and maladaptive preventive behaviors [17]. Notably, during the pandemic, knowledge and attitudes regarding infectious diseases were often associated with emotions like panic and anxiety, which impeded efforts to control disease transmission [10]. Consequently, understanding the current KAP level of the population regarding COVID-19 offers valuable insights for addressing the gaps in preventive strategies and health promotion programs [18,19,20]. The general objective of this study was to determine the knowledge, attitudes, and practices in a Colombian Caribbean population that had recovered from COVID-19 during the pandemic.

## 2. Materials and Methods

### 2.1. Setting

The Caribbean region of Colombia, located on the northern coast of the country and bordering Venezuela, Panama, and the Atlantic Ocean, is one of the most populated regions in the country and also one of the hardest hit by the COVID-19 pandemic. According to official figures, between 2020 and 2021, 2.4 million infections were reported in this area, which is equivalent to 40.2% of the national total. Among the causes of this high incidence are the high demographic density, the flow of people, and the socioeconomic conditions of the population [21].

### 2.2. Study Design

A descriptive cross-sectional study was conducted among the Colombian Caribbean population.

### 2.3. Participants

Participants were recruited using a non-probabilistic, non-discriminatory exponential snowball sampling method from July 2020, to August 2021. To disseminate our call for participation, we utilized social networking platforms such as Facebook, Instagram, and a ‘WhatsApp’ chat group. Invitations to participate in the survey, along with a link to a questionnaire, were shared from both our personal accounts and those of the research team members. These invitations were also shared by colleagues and other research groups working in various cities within the Colombian Caribbean region. The invitation conveyed essential information about the study, its objectives, and the estimated time required to complete the questionnaire. It also emphasized the importance of informed consent. To underscore the voluntary and confidential nature of participation, respondents were initially required to confirm their voluntary participation with a ‘yes’ or ‘no’. Upon confirmation, participants were directed to complete the self-reporting questionnaire. We monitored the number of responses by checking the ‘responses’ section of Microsoft Forms. The sample size was determined using the Epi Info Statcal module www.cdc.gov/epiinfo (accessed on 17 March 2020), considering an estimated population size of 1,248,000 individuals aged between 15 and 81 years. The study aimed for an expected frequency, margin of error, and confidence level of 50%, 5%, and 95%, respectively. Regarding inclusion criteria, the study encompassed individuals aged 15 years or older who had recovered from COVID-19 at least 3 months prior. A total of 350 participants were excluded from the study. This exclusion group comprised individuals under 15 years of age, healthcare professionals (due to their in-depth knowledge of the subject matter), and those who declined to provide informed consent online.

### 2.4. Study Variables

The study outcomes encompass individuals’ levels of knowledge, attitudes, and practices in relation to COVID-19. The dependent variables pertain to the demographic characteristics of the participants.

### 2.5. Data Sources/Measurement

The questionnaire consisted of a total of 24 questions, divided into two parts. The first part comprised 8 questions related to sociodemographic characteristics, while the second part consisted of the KAP questionnaire developed by Zhong et al., which featured 16 questions. Among these, 12 questions assessed the level of knowledge, 2 explored participants’ attitudes, and 2 examined their practices in relation to COVID-19 [22]. Of the 12 knowledge-related questions, 4 pertained to clinical manifestations, 4 addressed transmission routes, and 4 focused on prevention and control measures for COVID-19. The two attitude-related questions sought participants’ opinions on the final control of the virus and their confidence levels in the eradication of the virus. Participants answered the attitude-related questions with “true”, “false”, or a “Don’t know” option. Correct answers received 1 point, while incorrect or unknown responses were assigned 0 points. This resulted in a total knowledge score that ranged from 0 to 12, with a higher score indicating greater knowledge of COVID-19 [16].

To disseminate the questionnaire, we used social networking platforms (Facebook and Instagram) and a “WhatsApp” chat group. From our personal accounts and those of the members of the research team, we shared the link to the form with colleagues and other research groups working in the different cities of the Colombian Caribbean region. To monitor the number of responses, we consulted the “responses” section of the Microsoft Forms. A sample of the recovered COVID-19 population was randomly selected to avoid sampling bias.

### 2.6. Bias

To mitigate bias, the questionnaire was designed to maintain anonymity and to further ensure objectivity; the questions and response options were randomly reordered. This approach was adopted due to the sensitivity of the subject matter being addressed.

### 2.7. Statistical Analysis

In terms of statistical analysis, the data were first used to describe the study sample. Continuous variables were represented using the mean and standard deviation (SD) as measures of central tendency and dispersion, respectively. These values were employed to characterize the data distribution for comparisons with similar studies. Categorical variables were expressed as percentages. Differences between the groups for continuous variables were assessed using Student’s *t*-test, while the knowledge, attitudes, and practices’ variables were analyzed using Pearson’s chi-square test, with the significance level set at *p* < 0.05. Additionally, odds ratios and confidence intervals were calculated. The data analysis was carried out using SPSS v.25.

## 3. Results

A total of 1424 individuals willingly participated in the study. The demographic characteristics of the participants are summarized in Table 1 and Table 2.

The average age of the population was 23.1 years, with a standard deviation (SD) of 8.6. Of the participants, 819 (57.5%) were female. Educational attainment varied, with 25% having achieved a university-level of education, 36% completing secondary school, 29.4% holding a technical degree, and 2.6% having completed primary school. In terms of occupation, 12.7% were unemployed (this included housewives), 3.7% were not working due to health reasons related to the pandemic, 45% were students, and 49% were engaged in various occupations, including being pensioners, employees, and independent workers.

The mean overall correct response rate for each of the items related to COVID-19 knowledge differed significantly among gender, education level, and occupation (*p* < 0.001).

The average score for the 12 COVID-19 knowledge questions was 8.5, with a standard deviation of 2.1. This indicates an overall correct response rate of approximately 68.7% (8.5 × 12/100) among participants (as shown in Table 3).

When examining knowledge about the clinical manifestations of COVID-19, the rate of correct answers was notably higher, at 74.81% (*p* < 0.001). However, for knowledge related to transmission routes, the average correct response rate was lower, at 43.3% (*p* < 0.001). In contrast, knowledge about control and prevention measures had a higher mean score of 76.6% (*p* < 0.001).

Additionally, a significant portion of respondents held optimistic views regarding COVID-19 control. Specifically, 60.4% believed that COVID-19 would eventually be successfully controlled, and 70.9% expressed confidence in the ability of government agencies to effectively combat the virus. Overall, the general population displayed a positive attitude toward the potential success of COVID-19 control, with an attitude rating of 68.8% (*p* < 0.001).

In terms of the population’s attitudes towards COVID-19, Table 4 provides an overview of the variables on a scale with a maximum possible value of 100%.

Participants who lacked knowledge about COVID-19 and believed that it was not successfully controlled were found to be seven times more likely to become infected (95% CI: 3.6–13.6) compared to those who believed it would be controlled or those who were unaware of the topic, which was 901 individuals (69.7%). Similarly, participants who possessed knowledge about COVID-19 and believed that the recommended guidelines would successfully control the pandemic had a lower risk of infection, which was only 122 participants (92.4%).

Regarding the confidence of the surveyed population in the actions taken by the local administration, those with a low level of knowledge who lacked confidence were found to be 5.2 times more likely to be infected (95% CI: 2.7–10.1) than those with a low level of knowledge who did have confidence in the measures taken. These findings highlight the significant impact of knowledge and attitudes on the risk of COVID-19 infection.

Regarding knowledge and preventive practices, it is noteworthy that a large majority of participants, approximately 90%, reported going out to crowded places. Importantly, many of these individuals lacked sufficient knowledge about the clinical manifestations, transmission routes, and prevention and control measures related to COVID-19. Those who did go out to crowded places without adequate knowledge were found to be 1.7 times more at risk (95% CI: 1.0–2.8) than those who did not go out, as indicated in Table 5.

Additionally, among the participants, 526 individuals reported not using a mask in the last few days when leaving home. Remarkably, only 10.1% of these individuals were aware of the risks associated with not using a mask, while the majority, 89.9%, were unaware of these risks. This underscores the importance of education and awareness in promoting preventive practices during the COVID-19 pandemic.

## 4. Discussion

This study aimed to assess the level of knowledge, attitudes, and practices related to COVID-19 among a specific population in Colombia with a history of the disease, focusing on a three-month period. The study predominantly involved male participants with an educational background, and their socioeconomic level was low. The findings revealed that the overall rate of correct responses among participants was 68.7%, indicating that more than half of the respondents possessed some level of awareness about COVID-19. However, it is important to note that the study primarily examined basic aspects of COVID-19, such as symptoms, transmission routes, and prevention and control measures. Although the overall rate of correct responses to the 12 questions of the COVID-19 knowledge questionnaire was 68.7% among the participants, considering that a significant proportion of the participants were students and persons with high school or university levels of education, it would be expected that their levels of knowledge would be higher due to their education and the national strategies implemented in educational institutions during the pandemic.

These data confirm the view of Raghupathi V and Raghupathi W, who stated that people with higher academic degrees have better healthcare and longer life expectancies than those with lower educational backgrounds [23]. However, the relationship between a person’s academic degree and his or her ability to act to prevent COVID-19 transmission remains unclear [23]. This is consistent with the research conducted in Paraguay via a cross-sectional study in a population of young adults and university students, where they obtained an overall rate of correct answers of 62% [24]; this differs from the results found in the population of China [22], Ecuador [25], Tanzania [26], Peru [27], Brazil [28], and Bogotá [29], where the level of knowledge regarding COVID-19 was higher than in this study. These findings underscore the importance of continued education and awareness campaigns, particularly in populations with varying levels of education, to enhance knowledge and promote better preventive practices during the ongoing COVID-19 pandemic [29].

When examining specific areas of the population’s knowledge level regarding COVID-19, it becomes evident that the lowest level of knowledge was related to transmission routes, with a mean score of 43.3 and a standard deviation of 26.8. This suggests that participants had limited clarity about how they could become infected with the virus. In contrast, participants demonstrated a better understanding of the virus’ clinical manifestations, with a mean score of 74.81 and a standard deviation of 23.5. Additionally, they had a relatively strong grasp of the practices they should implement to prevent and control COVID-19, as indicated by a mean score of 76.6 and a standard deviation of 17. These results are similar to those obtained by a cross-sectional study conducted in China with 120,000 respondents, where gaps in knowledge, misconceptions, and discriminatory attitudes regarding COVID-19 [30] were found. Regarding the results in the areas of knowledge of this study, other investigations such as the one conducted in a population-based sample of Ecuador presented similar results; they showed that the lowest level of knowledge was that of the routes of transmission (66.1%), followed by knowledge regarding the symptoms of the virus (82.28%), while the population had a greater mastery of the prevention and control (87.52%) of COVID-19 [25]. However, the results of the study in China differ in terms of what was found, since the population reflected lower levels of knowledge concerning the clinical manifestations of the virus (83.45%), and higher levels on the routes of transmission (92.83%) and prevention and control (92.92%) [30]. These variations in knowledge levels across different regions highlight the importance of tailoring educational campaigns to address specific areas where knowledge is lacking, or misconceptions exist. In the case of this Colombian population, enhancing the understanding of transmission routes may be particularly important to improve the overall awareness of COVID-19.

In terms of preventive measures, it is noteworthy that most participants favored adopting barrier methods, which included using masks and gloves, practicing regular handwashing, and adhering to social distancing guidelines. These measures were considered essential for reducing the risk of infection [22]. In low-income countries, men, people under 30 years of age, and people with 12 years of education or less had the lowest practice scores. Practice scores were lower than 60% in Africa and Europe/Oceania. A significant positive correlation was observed between knowledge and practice, and attitude and practice [7]. Similarly, in Nicaragua, 98% of respondents knew the symptoms of COVID-19, 88% knew how it is transmitted, and 81% knew the prevention measures. However, only 54% of those who wore masks and 38% of those who practiced social distancing were unaware of the etiology, transmission mechanism, and symptoms of COVID-19, which may lead to underestimation of the risk, delay in seeking medical attention, and the use of ineffective or dangerous home remedies [11].

Among the respondents, 65.9% felt that COVID-19 would be successfully controlled, which was higher than in the study conducted in Ecuador (47.5%) in a mostly female population, aged 30–49 years, married, and with a university education [25]. However, this figure was lower than results from studies in Paraguay (66.28%) [24], China (90.08%) [22], and Tanzania (85%) [26]. Meanwhile, 71.8% of our participants were confident that the Colombian government would handle the COVID-19 health crisis well during the pandemic, a result much lower than those obtained from the Chinese (97.1%) and Tanzanian (96%) populations [22,26], in addition to being significantly higher than another study of a Colombian population [29], specifically of health workers in an health care provider institution (IPS) (52.7%) and higher than other research in South American countries such as Ecuador (63.5%), Peru (23.1%), Brazil (41%), Paraguay (86.71%), and Venezuela (42.8%) [25,27,31,32]. The participants’ optimism may be related to the government’s control measures even before the first reported case in the country in 2020. The administration began isolation and quarantine measures for travelers from countries with high infection rates, issued biosecurity protocols for each sector, implemented economic relief, declared a health alert, limited the movement of people and vehicles throughout the country, and closed borders, among other measures that may have provided confidence to people regarding the battle against the virus [29,33].

However, the strict prevention and control measures, including the total closure of establishments managed by the government, seemed insufficient during the survey. The practices employed by the Colombian population were not entirely cautious, with approximately 89.6% reporting having gone to crowded places. A similar scenario was reported in a cross-sectional study of 10,551 participants from 23 countries in Asia, Africa, Europe, and America; it was revealed that the KAP level towards COVID-19 was moderate, knowledge was positively associated with good practices, and this relationship was mediated by attitudinal factors (perceptions), Risk, and efficacy beliefs. The populations with low levels of knowledge, attitudes, and practices included men, the youngest, those with lower educational levels, the unemployed, those living in rural areas, and those residing in low- and middle-income countries [8].

Likewise, in countries like Venezuela (56.3%) and African countries [32,33,34,35], the population violated mandatory isolation by visiting crowded places for reasons such as having to return to work due to insufficient income and food shortage. However, this was not observed in studies from China (96.4%), Ecuador (97%), and Paraguay (88.35%) [22,25,28], where most respondents indicated that they had not visited crowded places. However, while most of this study’s participants reported visiting crowded places, they did take some preventive measures, such as using face masks (63.06%), albeit with a lower frequency than in the studies from Ecuador (93%), China (98%), Peru (98.6%), and Paraguay (74.31%) [25,28,30,31]. These results indicate a gap between adherence to theoretical knowledge and satisfactory practices [36].

The study revealed several interesting findings regarding the Colombian population’s attitudes and behaviors during the COVID-19 pandemic: a significant portion of the participants, 65.9%, believed that COVID-19 would eventually be successfully controlled, which was higher than the findings from a study in Ecuador (47.5%) [16] involving a predominantly female population aged 30–49 years with university education.

Approximately 71.8% of participants expressed confidence in the Colombian government’s handling of the COVID-19 health crisis during the pandemic, although this confidence level was lower than those observed in China (97.1%) and Tanzania (96%) [29,30]. In contrast, this confidence level was higher than that reported among Colombian health workers in a separate study (52.7%) [18] and higher compared to populations in South American countries such as Ecuador (63.5%), Peru (23.1%), Brazil (41%), Paraguay (86.71%), and Venezuela (42.8%) [25,27,28,31,32].

Despite the government’s implementation of strict prevention and control measures, including the total closure of government-managed establishments, a substantial portion of the Colombian population (approximately 89.6%) reported going to crowded places.

This behavior of visiting crowded places during the pandemic was not unique to Colombia and was also observed in Venezuela (56.3%) and certain African countries [29,32,34], where people violated mandatory isolation due to economic constraints and food shortages.

In contrast, studies in China (96.4%) [22], Ecuador (97%) [25], and Paraguay (88.35%) [28] reported lower rates of people visiting crowded places during the pandemic. However, despite visiting crowded places, many participants did take some preventive measures, such as using face masks (63.06%). Although this rate was lower than in some other countries, it still indicates a level of awareness and compliance with preventive practices. The study highlighted a gap between knowledge and practice among participants. While a majority had theoretical knowledge about COVID-19, their actual behaviors did not consistently align with this knowledge [20,31].

The results indicated that those with low knowledge about the disease and health measures, coupled with low confidence in the national government’s management of the crisis, were at a significantly higher risk of infection compared to those with low knowledge but higher confidence. This suggests the need for effective education and communication strategies that enhance the population’s knowledge and confidence in preventive measures while promoting compliance with recommended practices.

To address the knowledge and practice gap, it is recommended that tailored health education strategies be developed to cater to the specific characteristics and needs of different population groups.

For example, education programs targeting single female students between 19 and 30 years of age from lower socioeconomic backgrounds could be implemented. These programs should focus on improving attitudes and practices regarding COVID-19 and other future health issues to ultimately reduce infection and complication risks.

Such programs should emphasize evidence-based knowledge about COVID-19 symptoms, aftereffects, and prevention measures in accordance with scientific guidance and the health authorities’ recommendations.

In summary, this study’s findings highlight the importance of not only increasing knowledge but also translating that knowledge into practical, preventive behaviors. Tailored health education strategies, coupled with efforts to enhance confidence in government measures, can play a crucial role in promoting better compliance with recommended practices and reducing the risk of COVID-19 infection [37].

According to the CEPAL and UNESCO report, the gaps in the knowledge, attitudes, and practices surrounding COVID-19 in Latin America are due to several factors, such as the lack of reliable and accessible information, distrust in health authorities, the influence of social networks and the media, cultural and linguistic diversity, socioeconomic conditions, and geographical barriers [38].

In order to design personalized education campaigns, it is necessary to know what factors influence information-seeking behavior about COVID-19, allowing for health communication strategies to be adjusted to increase their effectiveness, equity, and impact on specific subgroups, including those most vulnerable to severe illness from COVID-19, as well as measure the campaign performance and results. The participation of the community and its social leaders in the solutions is essential to generate trust in the community and maintain progress in healthcare. Give priority to those who do not have access to information and communication technologies (ICT) or telecommunications or broadcasting services at home. Diversify online education alternatives according to the devices available in different contexts and their conditions of use. Strengthen focused, measured remote education that encourages recreational activities. Ensure the learning necessary to face the emergency with a framework of intersectoral collaboration. Health education is a key factor to the prevention and control the COVID-19 pandemic, but it requires greater coordination between the different actors and institutions involved. The implementation of educational programs adapted to local needs and contexts is recommended, as well as the promotion of a culture of truthful and responsible information among the population [36,39].

### Limitations

This survey was only conducted in the Caribbean region of Colombia, which has a different culture from the rest of the country. Therefore, there are limitations to generalizing these results.Since the study was conducted among individuals affected by COVID-19, there is a potential bias in that these individuals may have a higher level of knowledge than the rest of the population, which could affect the nature of the results. Additionally, as it is a cross-sectional study it is more vulnerable to bias.This study was conducted at the end of 2020 and the beginning of 2021, when attitudes and behaviors were not the same as at the beginning of the pandemic. Therefore, comparing these results with studies conducted at the beginning of the pandemic may not be the most appropriate.This study can only be compared with studies conducted during the same period in which it was conducted.The questionnaire used in this study was obtained from Zhong et al. [22], so it may be a questionnaire with simple but important questions. In the study, there were individuals with very low levels of education who may not have been able to comprehend a more complicated questionnaire due to their educational level.

## 5. Conclusions

The population studied, in general, maintained favorable attitudes towards the idea of being able to obtain adequate control of the spread of the virus, trusted the Colombian government’s measures, and were optimistic about the possible eradication of the virus. However, some areas were found in which improvements could have been made, such as outdoor activities, where a high level of attendance was reported in crowded places, and although the use of face masks was the main protective measure, it is also necessary to take preventive and educational measures to avoid this type of crowding.

On the other hand, to reduce the risk of infection and close the gap between different levels of knowledge about the control of COVID-19, it would have been beneficial to establish an interdisciplinary support network and carry out massive educational campaigns aimed at specific population groups, promoting good clinical practices that would generate greater awareness about COVID-19 and thus have been able to contribute to stopping its spread.

Addressing these areas would have helped to further empower the population to effectively prevent and control the spread of the virus and ultimately contributed to the collective effort to manage the pandemic.

In essence, and not only in the case of the COVID-19 pandemic, but in the presence of any other epidemic, there will always be the opportunity to improve the knowledge, attitudes, and practices related to self-care, especially when focusing mainly on increasing education, communication, and community participation.

## Figures and Tables

**Table 1 healthcare-11-03119-t001:** Demographic characteristics of the sample population compared with the Columbian and Columbian Caribbean populations.

Characteristics		All * [n (%)]	Caribbean * [n (%)]	Sample [n (%)]
Sex	Male	21,570,493 (48.8)	4,091,559 (48.7)	574 (44.4)
	Female	22,593,924 (52.1)	4,311,979 (51.3)	718 (55.6)
Age (Year)	15–19	3,852,255 (8.72)	942,893 (24.4)	498 (35)
	20–29	7,632,462 (17.28)	1,660,091 (21.8)	783 (55)
	30–44	9,311,837 (21.08)	1,971,041 (21.1)	100 (7.0)
	>45	12,959,848 (29.34)	2,542,130 (19.6)	43 (3.0)
Socioeconomic Level **	1 (very low)	10,466,966 (23.7)	2,025,251.21 (24.1)	380 (26.7)
	2 (low)	17,798,260 (40.3)	3,252,166.88 (38.7)	554 (38.9)
	3 (medium low)	11,217,761 (25.4)	1,420,196.91 (16.9)	369 (25.9)
	4 (medium)	3,268,166 (7.4)	907,581.46 (10.8)	92 (6.5)
	5 (medium high)	927,452 (2.1)	521,018.98 (6.2)	18 (1.3)
	6 (high)	485,808 (1.1)	277,316.26 (3.3)	11 (0.8)
Educational Level	None	1,806,365 (0.8)	639,802 (7.6)	3 (2)
	Primary	4,893,091 (9.1)	988,481 (11.7)	37 (2.6)
	Secondary	1,614,994 (34.1)	366,124 (4.3)	513 (36)
	Technical	2,319,506 (36.4)	555,573 (6.6)	419 (29.4)
	Professional	4,191,249 (18.9)	725,487 (8.6)	359 (25.2)
	Specialist, Master, Doctor	1,137,777 (0.8)	132,985 (1.5)	93 (6.5)
Marital Status	Single	20,668,947 (46.8)	4,068,401.83 (47.9)	916 (64.3)
	Married/Common	11,438,584 (25.9)	2,106,395.94 (24.8)	209 (14.7)
	law marriage	10,025,322 (22.7)	1,885,564.1(22.2)	226 (15.9)
	Widow(er)	529,973 (1.2)	127,402.98 (1.5)	19 (1.3)
	Divorced/Separated	1,501,590 (3.4)	305,767.15(3.6)	54 (3.8)
Occupation	Unemployed (included housewives)	10 (7.6)	78 (6.0)	88 (6.2)
	Unemployed due to pandemic/health	3 (2.3)	49 (3.8)	52 (3.7)
	Student	47 (35.6)	598 (46.3)	645 (45.3)
	Pension	8 (6.1)	16 (1.2)	24 (1.7)
	Independent	34 (25.8)	179 (13.9)	213 (15.0)
	Salaried	30 (22.7)	372 (28.8)	402 (28.2)

Source: * DANE (National Administrative Department of Statistics) was used for demographics and population. https://www.dane.gov.co/index.php/estadisticas-por-tema/demografia-y-poblacion (accessed on 24 November 2023). ** Colombian socioeconomic level on https://www.dane.gov.co/files/geoestadistica/Preguntas_frecuentes_estratificacion.pdf (accessed on 24 November 2023). According to DANE, the population over 15 years of age in the Caribbean region is estimated to be 8,403,532 people for the year 2021, which is 21.9% of the population over 15 years of age in Colombia. This figure is based on the population projections that DANE makes based on the results of the 2018 National Population and Housing Census, which was the last census carried out in Colombia. Of this population, 48.7% are men and 51.3% are women, that is, there are 4,091,559 men and 4,311,973 women aged 15 years or over in the Caribbean region.

**Table 2 healthcare-11-03119-t002:** Demographic characteristics of participants and COVID-19 knowledge percent in terms of demographic variables.

Characteristics		Knowledge [n (%)]		*p* Value *
		Yes	No	
Sex	Male	31 (23.5)	574 (44.4)	0.01
	Female	101 (76.5)	718 (55.6)	
Age (Year)	15–18	21 (15.9)	260 (20.1)	0.06
	19–30	97 (73.5)	913 (79.7)	
	31–43	4 (3.0)	84 (6.5)	
	>44	10 (7.6)	35 (2.7)	
Socioeconomic Level **	1 (very low)	33 (25.0)	347 (26.9)	0.22
	2 (low)	55 (41.7)	499 (38.6)	
	3 (medium low)	40 (30.3)	329 (25.5)	
	4 (medium)	3 (2.3)	89 (6.9)	
	5 (medium high)	1 (0.8)	17 (1.3)	
	6 (high)	0 (0.0)	11 (0.9)	
Education Level	None	1 (0.8)	2 (0.2)	0.01
	Primary	12 (9.1)	25 (1.9)	
	Secondary	45 (34.1)	468 (36.2)	
	Technical	48 (36.4)	371 (28.7)	
	Professional	25 (18.9)	334 (25.9)	
	Specialist	1 (0.8)	60 (4.6)	
	Master	0 (0.0)	30 (2.3)	
	Doctor	0 (0.0)	2 (0.2)	
Marital Status	Single	84 (63.6)	832 (64.4)	0.17
	Married/Common law marriage	36 (27.3)	399 (30.9)	
	Widow(er)	3 (2.3)	16 (1.2)	
	Divorced/Separated	9 (6.8)	45 (3,5)	
Occupation	Unemployed (included housewives)	10 (7.6)	78 (6.0)	0.01
	Unemployed due to pandemic/health	3 (2.3)	49 (3.8)	
	Student	47 (35.6)	598 (46.3)	
	Pension	8 (6.1)	16 (1.2)	
	Independent	34 (25.8)	179 (13.9)	
	Salaried	30 (22.7)	372 (28.8)	

* Student’s *t*-test, significance level *p* < 0.05. ** Colombian socioeconomic level on https://www.dane.gov.co/files/geoestadistica/Preguntas_frecuentes_estratificacion.pdf (accessed on 9 October 2023).

**Table 3 healthcare-11-03119-t003:** Average number of correct answers from the participants for the knowledge, attitudes, and practices questions.

Knowledge, Attitudes, and Practices	Mean (SD)	*p* *
Clinical manifestations	74.81 (23.5)	0.01
Transmission routes	43.3 (26.8)	0.01
Control and prevention measures	76.6 (17.0)	0.01
Attitudes toward control and trust related to COVID-19	60.4 (36.8)	0.01
Practices	70.9 (34.0)	0.01

***** Student’s *t*-test, significance level *p* < 0.05.

**Table 4 healthcare-11-03119-t004:** Analysis of the relationship between knowledge and attitudes toward COVID-19.

Attitudes		Knowledge [n, (%)]		OR ^1^ (95% CI)	*p* *
		Yes	No		
Do you agree that COVID-19 will eventually be successfully controlled?	No	475 (36.8)	10 (7.6)	7.0 (3.6–13.6)	0.01
	Yes	817 (63.2)	122 (92.4)		
Are you confident that the local administration is handling the COVID-19 health crisis very well?	No	391 (30.3)	10 (7.6)	5.2 (2.7–10.1)	0.01
	Yes	901 (69.7)	122 (92.4)		

^1^ OR: odds ratios, 95% CI = 95% confidence interval, ***** significance level *p* < 0.05.

**Table 5 healthcare-11-03119-t005:** Analysis of the relationship between knowledge and practices against COVID-19.

Practices		Knowledge [n, (%)]		OR ^1^ (95% CI)	*p* *
		Yes	No		
In the last few days, have you gone anywhere crowded?	No	1165 (90.2)	111 (84.1)	1.7 (1.0–2.8)	0.02
	Yes	127 (9.8)	21 (10.4)		
In recent days, have you worn a mask when leaving the house?	No	473 (36.6)	53 (40.2)	0.8 (0.5–1.2)	0.4
	Yes	819 (63.4)	79 (59.8)		

^1^ OR: odds ratios, 95% CI = 95% confidence interval, * significance level *p* < 0.05.

## Data Availability

Data are available from the corresponding author on reasonable request. They are not publicly available as further sub-analyses are currently ongoing.

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
