# Peer review of "The Level of Knowledge, Attitudes, and Practices in a Caribbean Colombian Population That Recovered from COVID-19 during the Pandemic"

_healthcare, 2023, doi:10.3390/healthcare11243119_

Round 1

Reviewer 1 Report

Comments and Suggestions for Authors

Thank you for requesting my review of this manuscript. Here are some critical review comments and suggestions that could help improve the quality of this work:

1. The study provides valuable insights into the knowledge, attitudes and practices related to COVID-19 in a specific population in Colombia. However, the manuscript would benefit from more clearly situating the study and findings within the broader literature on this topic globally and in Latin America.

2. The authors could expand the introduction to provide more background on KAP studies related to pandemics and infectious disease outbreaks, and how findings have informed public health responses. This would help frame the rationale and significance of the study.

3. In the discussion and conclusion, the authors could relate their findings more directly to the existing evidence base and identify how their results compare or contrast with KAP studies in other regions and populations. This would strengthen the interpretation and implications of the study.

4. The manuscript would be strengthened by providing more details on the survey methodology and sampling approach, as well as discussing any limitations related to sample representativeness and generalizability of the findings.

5. The authors could consider shortening some lengthy sentences for clarity. The language could also be polished for grammar, spelling and consistent terminology.

Specific Comments:

1. In the abstract conclusion, rephrase "it is crucial to establish an interdisciplinary support network" for clarity. Consider changing to "establishing an interdisciplinary support network is crucial".

2. In the Introduction, expand on the evidence that compliance with control measures relied heavily on KAP levels among citizens. Cite relevant studies.

3. In the Discussion, relate the findings on knowledge levels more directly to the results of KAP studies in other regions mentioned. Elaborate on the implications.

4. In the Discussion, expand on the recommendations for tailored health education campaigns. Provide more specifics on how these could be designed and targeted.

5. In the Conclusion, consider reorganizing the paragraph starting "In essence..." for better flow. Move the sentence on "addressing these areas" earlier.

6. Carefully review usage of terms like "practices", "behaviors", "measures" for consistency. Clarify if certain terms are meant to refer to specific concepts.

7. Review references to ensure journal name abbreviations and formatting are consistent with journal guidelines.

I hope these suggestions are helpful for the authors in revising this manuscript. 

Comments on the Quality of English Language

Syntax errors. I am requesting to check the manuscript for typographical and syntax errors. 

Author Response

We greatly appreciate the comments and review made to our manuscript. We responded to all the comments made by you, these were made in the manuscript.

Thank you for requesting my review of this manuscript. Here are some critical review comments and suggestions that could help improve the quality of this work:

  1. The study provides valuable insights into the knowledge, attitudes and practices related to COVID-19 in a specific population in Colombia. However, the manuscript would benefit from more clearly situating the study and findings within the broader literature on this topic globally and in Latin America.

Response: We include new references globally and in Latin America. With this, the study is placed in a better context worldwide and in Latin America. The study was located more broadly worldwide and in Latin America

  1. The authors could expand the introduction to provide more background on KAP studies related to pandemics and infectious disease outbreaks, and how findings have informed public health responses. This would help frame the rationale and significance of the study.

Response: The introduction was expanded and several KAP studies were discussed and included.

  1. In the discussion and conclusion, the authors could relate their findings more directly to the existing evidence base and identify how their results compare or contrast with KAP studies in other regions and populations. This would strengthen the interpretation and implications of the study.

Response: The discussion and conclusion were corrected, and our results were contrasted with the KAP studies in other regions and populations.

  1. The manuscript would be strengthened by providing more details on the survey methodology and sampling approach, as well as discussing any limitations related to sample representativeness and generalizability of the findings.

Response: The methodology of how the survey was carried out and details of the sampling were written. All limitations of the study were included in a limitations section.

  1. The authors could consider shortening some lengthy sentences for clarity. The language could also be polished for grammar, spelling, and consistent terminology.

Response: Sentences were shortened, and the manuscript was again checked for grammar.

Regarding the specific comments, all were made from 1 to 7, considering all the recommendations made (see manuscript, please)

Specific Comments:

  1. In the abstract conclusion, rephrase "it is crucial to establish an interdisciplinary support network" for clarity. Consider changing to "establishing an interdisciplinary support network is crucial".

Response: Suggested changes were made, rephrasing "it is crucial to establish an interdisciplinary support network" to "establishing an interdisciplinary support network is crucial".

  1. In the Introduction, expand on the evidence that compliance with control measures relied heavily on KAP levels among citizens. Cite relevant studies.

Response: Expanded the evidence in the introduction that compliance with control measures was highly dependent on the levels of KAP among citizens. added citations (6-9)

  1. In the Discussion, relate the findings on knowledge levels more directly to the results of KAP studies in other regions mentioned. Elaborate on the implications.

Response: The findings of the study on knowledge levels were related to the results of KAP studies in other regions. The following citations were added (21,24, 38-40)

  1. In the Discussion, expand on the recommendations for tailored health education campaigns. Provide more specifics on how these could be designed and targeted.

Response: The guidelines were detailed, and the design of personalized health education campaigns was discussed in depth. Citations (38-40)

  1. In the Conclusion, consider reorganizing the paragraph starting "In essence..." for better flow. Move the sentence to "addressing these areas" earlier.

Response: the suggested changes were made in the conclusion.

  1. Carefully review the usage of terms like "practices", "behaviors", and "measures" for consistency. Clarify if certain terms are meant to refer to specific concepts.

Response: we carefully reviewed these terms and clarified all of them.

  1. Review references to ensure journal name abbreviations and formatting are consistent with journal guidelines.

Response: The abbreviations and formatting of journal names were checked for compliance with journal guidelines.

I hope these suggestions are helpful for the authors in revising this manuscript. 

Response:  We greatly appreciate the comments and review made to our manuscript. We responded to all the comments made by you, these were made in the manuscript.

Reviewer 2 Report

Comments and Suggestions for Authors

I would thank the authors for his manuscript. After reviewing, I found some concerns that bothered me:

Fist, the study was conducted among individuals who have been affected by COVID-19. Thus their results could be taken carefully. Generally, infected individuals could have a high level of knowledge about some chapter of the disease which cold biases the results. Their attitude could also change affecting the nature of the results. You should take all these point in consideration.

The second point is related to the period of study. Thus the level of knowledge in late 2020 and the beginning of 2021 tend to be logically higher than the level of knowledge during the beginning of the pandemic. The attitudes and behaviours have also substantially changed after one year of the pandemic. Thus you should be very careful when comparing your results with the previous polished ones.

Third: The nature of the questions makes questionnaire very simplistic in 2020-2021 (after one year of the pandemic) especially for individuals who were affected by the COVID-19. Also, comparing your results (level of knowledge and attitudes) with other studies that were not taken in the same period and with different questionnaire is misleading that should be well discussed.

I have also other comments and remarks:

The title should included the affected (recovered) population.

Include the study period in the abstract.

Reformulate the sentence : "Lessons…uncertainty [5]." (Lines 51-54).

Enrich the introduction with previous related studies in the country and around the world (what was done before?) to define the hypothesis of the work.

Add the objectives of the study at the end f the introduction.

Results:

Line 140: delete : " and 819 (57.5%) were men".

Line 145: housewives are not considered as unemployed?

Line 149-150: delete this sentence (it y be in the discussion).

Table 1: what do you mean by known and unknown ?? (this did not mean Low and high level of knowledge??). (the same remark applies for all tables).

Add a column to describe the demographic characteristics of the population (n and %) before the knowledge %.

In the title of the same table: this may be he % of knowledge not the scores.

What do the numbers mean in the socioeconomic level?

You should indicate the used test (in the table footnotes) and the significant result in each table.

The table should be reorganized and placed just after their citation in the text.

Discussion

The discussion should be revised and reorganized. In addition to multiple self-statements, it contains multiple non referenced sentences and paragraphs even you cited the works o other researchers   (lines 234-237, 237-240, 280-291, 292-303, 304…).

Delete the sentence of line 201-202 ("The study…below 3"), and reformulate and summarize all this paragraph.

Revise the style of citation in lines 256 and 262.

Delete the subtitles and reformulate the discussion

Add the limitations of the study

Lines 207 and 232: what do you mean by acceptable? What is your scale?.

Revise the conclusion according to the revised results and discussion

Revise your list of reference by deleting the colors.

Comments on the Quality of English Language

Minor editing of English language required

Author Response

We greatly appreciate the comments and review made to our manuscript. We responded to all the comments made by you, these were made in the manuscript.

Author’s response to Reviewer 2:

We greatly appreciate the comments and review made to our manuscript. We responded to all the comments made by you, these were made in the manuscript.

  1.  the study was conducted among individuals who have been affected by COVID-19. Thus, their results could be taken carefully. Generally, infected individuals could have a high level of knowledge about some chapter of the disease which could bias the results. Their attitude could also change affecting the nature of the results. You should take all these points in consideration.

Response:  We discuss and include these aspects in a limitations section (Lines 640 to 657).

  1. The second point is related to the period of study. Thus, the level of knowledge in late 2020 and the beginning of 2021 tends to be logically higher than the level of knowledge during the beginning of the pandemic. The attitudes and behaviors have also substantially changed after one year of the pandemic. Thus, you should be very careful when comparing your results with the previous polished ones

Response:  We discuss and include these aspects in a limitations section (Lines 640 to 657).

  1. The nature of the questions makes questionnaire very simplistic in 2020-2021 (after one year of the pandemic) especially for individuals who were affected by the COVID-19. Also, comparing your results (level of knowledge and attitudes) with other studies that were not taken in the same period and with different questionnaire is misleading that should be well discussed.

Response: We discuss and include these aspects in a limitations section (Lines 640 to 657).

I have also other comments and remarks:

  1. The title should included the affected (recovered) population.

Response:  The title was adjusted in accordance with the recommendation (see the title)

  1. Include the study period in the abstract.

Response:  the study period was included in the summary, as recommended (Lines 15-16)

  1. Reformulate the sentence : "Lessons…uncertainty [5]." (Lines 51-54).

Response:  the suggested changes were made in these lines (lines 62-69)

  1. Enrich the introduction with previous related studies in the country and around the world (what was done before?) to define the hypothesis of the work.

Response:  Previous studies were added to the introduction, as recommended. Added citations (5, 7-10)

  1. Add the objectives of the study at the end of the introduction.

Response:  the general objective of the study was added at the end of the introduction as recommended (lines 123-125)

 Results:

  1. Line 140: delete : " and 819 (57.5%) were men".

Response:  was deleted as suggested and changed (lines 227)

  1. Line 145: housewives are not considered as unemployed?

Response:  housewives, were included in the group of unemployed people ( see table 1).

  1. Line 149-150: delete this sentence (it y be in the discussion).

Response:  the sentence was deleted

  1. Table 1: what do you mean by known and unknown?? (this did not mean Low and high level of knowledge??). (the same remark applies for all tables).

Response:  Changed "known/unknown" to knowledge/No knowledge (see table 1)

  1. Add a column to describe the demographic characteristics of the population (n and %) before the knowledge %.

Response:  The recommended modifications were implemented. ( n and % were included see table 1)

  1. In the title of the same table: this may be he % of knowledge not the scores.

Response:  The proposed suggestions were applied (see table 1)

  1. What do the numbers mean in the socioeconomic level?

Response:  Colombian Socioeconomic status can be measured qualitatively and quantitatively. The quantitative form was presented in the table and the change was made to the qualitative form that divides households into six strata: low-low (1), low (2), medium-low (3), medium (4), medium-high (5) and high (6). (reference of this was included see table 1)

  1. You should indicate the used test (in the table footnotes) and the significant result in each table.

Response:  The suggested changes were made. (we included the statistical test applied and the p significance)

  1. The table should be reorganized and placed just after their citation in the text.

Response:  The recommended modifications were implemented in the table

Discussion

18.The discussion should be revised and reorganized. In addition to multiple self-statements, it contains multiple non-referenced sentences and paragraphs even you cited the works o other researchers   (lines 234-237, 237-240, 280-291, 292-303, 304…).

Response:  The discussion was reviewed and reorganized (see lines 339-635).

  1. Delete the sentence of line 201-202 ("The study…below 3"), and reformulate and summarize all this paragraph.

Response:  The recommended modifications were applied, and the entire section was reformulated and summarized. (lines 341-342)

  1. Revise the style of citation in lines 256 and 262.

Response:  The recommended modifications were applied, and the citation style was revised for these lines

  1. Delete the subtitles and reformulate the discussion

Response:  All subtitles were removed, and the discussion was reformulated (lines 339-635).

  1. Add the limitations of the study

Response:  The limitations section of the study was introduced (see lines 640-657).

  1. Lines 207 and 232: what do you mean by acceptable? What is your scale?

Response:  The Questionnaire of Knowledge, Attitudes, and Practice towards COVID-19, assigns 1 point to a correct answer and 0 points to an incorrect/unknown answer. The total knowledge score ranges from 0 to 12, with a higher score denoting better knowledge of COVID-19 [22].

  1. Revise the conclusion according to the revised results and discussion

Response:  We reviewed the conclusion according to the revised results (lines 660-681)

Revise your list of reference by deleting the colors.

Response:  we review all references (lines 700-812)

Reviewer 3 Report

Comments and Suggestions for Authors

In the manuscript entitled “Level of Knowledge, Attitude, and Practices about COVID-19 2 in a Colombian Population during the Pandemic”, the authors conducted a descriptive cross-sectional study during the pandemic using a survey focused on knowledge, attitudes, and practices regarding COVID-19 pandemic. The authors highlight the importance of increasing knowledge as well as translating this knowledge into practical, preventive behaviors.

Although, the authors present their study in a scientific manner, cross-sectional studies are very vulnerable to bias.

Apart from this, I am attaching my comments for improving the manuscript.

I am recommending that the manuscript may be accepted for publication after major revisions.

Reviewer’s Comment:

1. The authors discussed the results in a proficient manner. But, it is a known fact that knowledge about transmission of an infection during a pandemic situation is very effective tool. Therefore, what is the novelty of this study?

2. What are the main limitations of this cross-sectional study?

3. What strategies are taken by the authors to avoid bias in sampling?

Author Response

We greatly appreciate the comments and review made to our manuscript. We responded to all the comments made by you, these were made in the manuscript.

1.The authors discussed the results in a proficient manner. But, it is a known fact that knowledge about transmission of an infection during a pandemic situation is very effective tool. Therefore, what is the novelty of this study?

Response:  The novelty of this study is that it evaluates the relationship between the knowledge, attitudes, and practices of the population regarding COVID-19 in a Colombian city and that it found that there was an important gap between the level of knowledge and compliance with preventive measures. The study also showed that gender, educational level, and occupation influence knowledge and attitudes about COVID-19, and that trust in health authorities and perception of success in controlling the pandemic are associated with a lower risk of contagion and transmission.

  1. What are the main limitations of this cross-sectional study?

Response:  We did a limitations section in which we discussed the main limitations of this cross-sectional study

  1. What strategies are taken by the authors to avoid bias in sampling?

Response: In the methods section, we included the participants' selection. This selection for all participants was random. (see lines 170-172)

Round 2

Reviewer 1 Report

Comments and Suggestions for Authors

I have no further comments to disclose. 

Comments on the Quality of English Language

Minor language editing is required, specifically syntax and punctions.

Author Response

Thank you very much for taking the time to review this manuscript.  Language editing was reviewed, specifically syntax and punctuation.

Reviewer 3 Report

Comments and Suggestions for Authors

The authors carefully and thoroughly revised their manuscript.

Thus, I am recommending that the manuscript may be accepted for publication.

Author Response

Dear  reviewer,

I thank you for the time you have taken to read and evaluate my manuscript. His feedback was very valuable to me and helped me improve the quality and clarity of my work.